# Vehicle Load Identification on Orthotropic Steel Box Beam Bridge Based on the Strain Response Area

**Jun-He Zhu, Chao Wang \*, Tian-Yu Qi and Zhuo-Sheng Zhou**

School of Civil, Architecture and Environment, Hubei University of Technology, Wuhan 430068, China
* Correspondence: cwang@hbut.edu.cn

**Abstract:** With the development of the economy and the rapid increase in traffic volume, an overload phenomenon often occurs. This paper studied a vehicle load identification technique based on orthotropic bridge deck stress monitoring data. The strain responses on the lower edge of multiple U-ribs were collected under vehicles crossed the deck. Firstly, an index based on the cross-correlation function of strain response between different measurement points on the same U-rib was used to evaluate vehicle speed. Secondly, a cosine similarity index was proposed to locate the transverse position of the vehicle. Finally, the unknown vehicle load was identified on the basis of a calibrated strain response area matrix. The effectiveness and anti-noise performance of the proposed method were verified using numerical simulation. An experimental model was designed and some strain gauges were installed to measure the strain response, and the test was carried out to further verify the algorithm's performance. Numerical and experimental results show that the proposed method could effectively identify the vehicle load with good anti-noise performance. Moreover, a calibration space was provided to guide practical engineering applications. The proposed method does not damage bridge decks, does not affect traffic, and is economical.

**Keywords:** load identification; strain response area; calibration spacing; cosine similarity; orthotropic steel deck



## 1. Introduction

With the rapid development of the economy, highway freight volume has followed a rapid growth, resulting in a traffic overload phenomenon becoming increasingly common. Overload will not only shorten the service life of bridges, but also significantly increase the probability of traffic accidents; severe threats to the safety of bridge structures may even be caused. The accurate identification of overload loads has important theoretical significance and engineering application value for structural health monitoring and safety assessment [1,2]. To accurately and stably identify the traffic load distribution on the bridge deck, Ge et al. [3] proposed an improved full-bridge traffic load monitoring method based on the YOLO-v3 convolutional neural network. The method adopted a training dual-target detection model to identify profiles of the vehicle and correct vehicle centroid for accurate vehicle location. Alberto et al. [4] proposed a computer vision-based method that simultaneously identifies the moving vehicle's load, the location of the vehicle on a bridge, and the displacement response, which are then used to update the finite element model in the monitored structural system. Currently, the main bridge vehicle load identification methods include dynamic response-based load identification, pavement weight-in-motion (WIM), and non-pavement bridge weight-in-motion (denoted as B-WIM). The dynamic response-based method identifies the vehicle load by solving the interaction force at the contact point between the vehicle and the bridge at any moment, through the vibration equation of the vehicle-bridge coupling system.

This method's application is susceptible to many factors, such as pavement unevenness and vehicle speed; furthermore, the algorithm is very complex, with many system

parameters [5,6]. Pavement WIM requires sensors to be buried by excavating the pavement, which affects traffic. Moreover, the durability of the sensors is poor, due to long-term direct vehicle rolling [7]. B-WIM technology involves installing sensors at the bottom of the bridge to monitor and identify vehicle loads, such as for dynamic strain-based B-WIM systems. The installation and maintenance of the sensors are easy, low-cost, and do not damage the pavement [8,9]. Thus, it is an up-and-coming load identification technology. The orthotropic deck steel box beam bridge with obvious local stresses and short influence lines is suitable for vehicle load identification using B-WIM technology [10,11]. Ojio and Yamada [12] proposed that the ratio of strain response areas of different vehicles driving across the bridge equals the ratio of the total vehicle weight. Chen et al. [13] also studied the correspondence between the strain response area and axle weight of large-span continuous steel truss bridge structures, in order to identify the gross weight of vehicles. Zhang et al. [14] proposed a dynamic iteration algorithm between Moses' algorithm and the measured influence line algorithm to correct the ill-conditioned problem of solving the equation. Zhao [15] used calibrated influence lines on site to calculate the axle weight, and he studied the influence of the choice of influence lines, the shape of the influence lines, and the sampling frequency on axle weight identification. These methods are primarily used for simply supported beam bridges, slab bridges, steel girder bridges, and truss bridges, and do not consider the impact of the transverse position of vehicles, i.e., the one-dimensional B-WIM system. Due to the prominent local force in the orthotropic deck, a change in a vehicle's transverse position on the bridge greatly influences load identification; thus, an analytical method needs to locate the transverse position of the vehicle in advance.

Quilligan [16] studied the two-dimensional B-WIM algorithm based on the influence surface and conducted experimental studies on integral slab bridges and girder bridges, respectively. Zhao [17] proposed an improved two-dimensional B-WIM algorithm for simply supported T-beam bridges, which considers the transverse distribution of the vehicle load on each main girder and calibrates the influence line for each main girder separately on site. A field experiment validated the effects of the method. Zhang [18] proposed a virtual monitoring method for the fatigue evaluation of the orthotropic steel deck. It was found that using the influence surface to identify the vehicle load could improve the recognition accuracy. Ma [19] identified the axle weight on the basis of strain variation in multiple U ribs when the vehicle load acted on different transverse positions.

When identifying the vehicle load on an orthotropic steel box beam bridge based on the measured dynamic strain, a critical problem is the location of the transverse position of the vehicle on the bridge deck, as well as the calibration of the strain response. The selection of the calibration spacing and the number of tests can greatly influence the identification results. This study analyzed the vehicle load identification method based on the strain response area. An index based on the cosine similarity was proposed to locate the transverse position of the vehicle. The influence of calibration spacing on gross vehicle weight identification is discussed, and the effectiveness and accuracy of the proposed algorithm are validated with numerical simulation and experimental tests.

## 2. Identification Method

The orthotropic steel bridge deck is directly subjected to wheel loads, and the local effects dominate its structural stresses. Under the support of the diaphragms, the stress status in the U-ribs is similar to that for a multi-span continuous beam; moreover, its stress influence line is very short. When a vehicle passes, the gross weight of the vehicle is related to the total area of the strain response of the U-rib and the transverse position of the vehicle. Firstly, a calibration vehicle is driven through the bridge, the strain response is measured, and the vehicle's speed is evaluated. Then, the calibrated strain response area can be calculated and the transverse position of vehicle can be located. In the following, the unknown vehicle is tested, and the practical strain response area and transverse position can be determined. Based on the calibrated strain response area at corresponding

transverse positions, the gross vehicle weight can be identified. The whole flowchart of the identification method is shown in Figure 1.

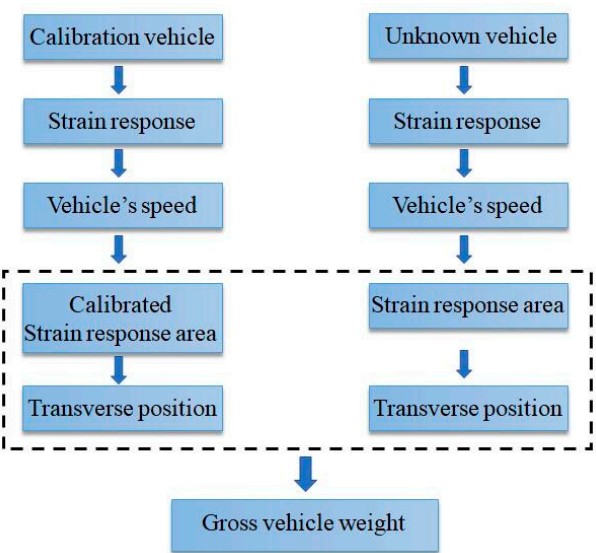

**Figure 1.** The whole flowchart of the identification method.

### 2.1. Identification of Vehicle Speed

For the convenience of explanation, a segmental model of an orthotropic bridge deck is selected as an example, as shown in Figure 2. Two strain gauges are installed at the lower edge of the same U-rib (named as N7 and N8) to measure the structural response when the vehicle across the deck. The longitudinal positions of the sensors are arranged in the adjacent segmental span (B-B and C-C sections in the figure), and the distance between measurement points is *d*.

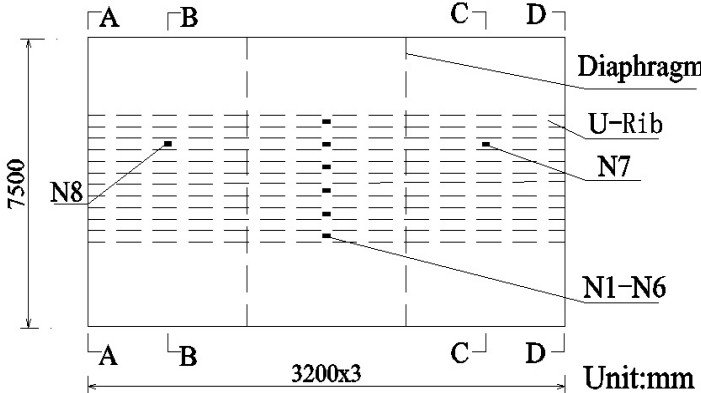

**Figure 2.** The segmental model of an orthotropic bridge deck and layout of the measurement points.

Assuming that the vehicle passes across the bridge deck at a uniform speed, the strain responses measured by the two measurement points are $P_7(t)$ and $P_8(t)$, and the sampling frequency is $f_s$. As a result of the local stress characteristic at the measurement points, their strain responses are similar, and the cross-correlation function can be calculated as follows:

$$R(\tau) = \int P_7(t) P_8(t + \tau) dt, \tag{1}$$

where $R(\tau)$ is the correlation function of the response of measurement points N7 and N8, $\tau$ is the time shift, and the extreme value of $R(\tau)$ corresponding to $\tau_{\max}$ is the time difference between the vehicle passing the two measurement points, as shown in Figure 3.

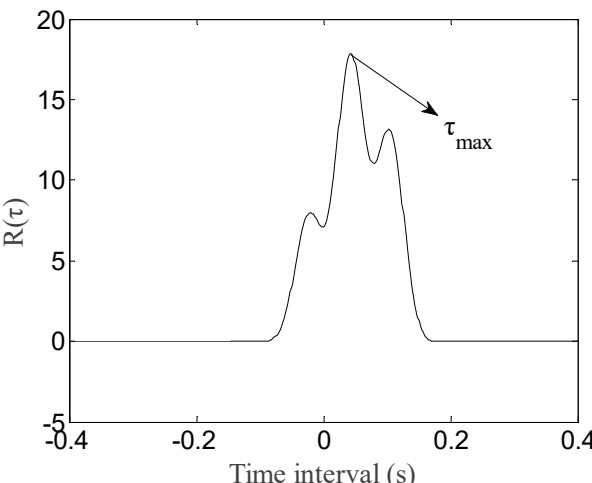

**Figure 3.** Cross-correlation function of measurement points N7 and N8.

The vehicle's speed can be estimated by the following equation:

$$v = \frac{d \times f_s}{\tau_{\max}},$$ (2)

It should be noted that the vehicle speed can be estimated directly based on the difference between x coordinates for the peak of N7 and N8. However, the peak is susceptible to noise interference and the time of peak is unstable. It is inaccurate to directly estimate the vehicle speed based on peaks. The proposed cross-correlation function determines the speed by analyzing two entire responses instead of a few peak points, which make the results more sable and ensure better anti-noise performance.

### 2.2. Identification of the Transverse Position

As shown in Figure 1, stress measurement points are placed on the lower edges of multiple U-ribs within a lane, and the actual number of measurement points varies with the lane width and the U-rib size. Without a loss of generality, six stress measurement points are placed here (N1 to N6), and the strain response of all measurement points is collected when the vehicle drives over the segmental model at different transverse positions. Due to the local force characteristics of the orthotropic bridge deck, the length of strain influence line of the measurement points on the U-rib is approximately the spacing between the front span and the back span of the diaphragm; thus, the entire strain response when the vehicle travels on the three-span spacing of the diaphragm is considered. Meanwhile, the response beyond the range is very small and has little effect on the following analysis. Firstly, the area of the strain response for each measurement point can be calculated based on the estimated vehicle speed, as shown in the following equation:

$$S_{ij} = v \times \int_{t_1}^{t_2} \varepsilon_{ij}(t)dt,$$ (3)

where $\varepsilon_{ij}$ is the strain response of measurement point $j$ when the vehicle passes the transverse position $i$; $S_{ij}$ is the area of the strain response $\varepsilon_{ij}$; $t_1$ is the moment when the first axle of the vehicle travels to the start point of the strain influence line (section A-A, as shown in Figure 2); and $t_2$ is the moment when the last axle of the vehicle travels to the end point of the strain influence line (section D-D, as shown in Figure 2).

For the transverse position $i$ of a vehicle through the model, the strain response area of all measurement points constitutes a vector $\mathbf{B}_i = [S_{i1}, S_{i2}, \cdots, S_{i6}]$, which will vary with the transverse position $i$ of the vehicle. Before identifying the vehicle's gross weight, the strain response area vectors under vehicle load at different transverse positions

are calibrated to obtain the standard strain response area matrix $\mathbf{B}^C = [\mathbf{B}_1^C, \mathbf{B}_2^C, \cdots, \mathbf{B}_n^C]$, where $n$ is the number of transverse positions. We can obtain the vectors in the strain response area at refined transverse positions by interpolating between the strain response areas at different positions.

Then, the strain response area vector $\mathbf{B}_T = [S_{T1}, S_{T2}, \cdots, S_{T6}]$ is measured under a vehicle with an unknown transverse position. A cosine similarity $CI$ index is introduced to estimate the transverse position by comparing vector $\mathbf{B}_T$ with the standard value $\mathbf{B}_i^C$, and is expressed as follows:

$$CI(i) = \cos(\mathbf{B}_T \cdot \mathbf{B}_i^C) = \frac{\mathbf{B}_T \cdot \mathbf{B}_i^C}{|\mathbf{B}_T| \cdot |\mathbf{B}_i^C|}, \tag{4}$$

The closer the $CI$ is to 1, the more similar the two vectors are. By finding the vector $\mathbf{B}_k^C$ when the $CI$ index is closest to 1, we can estimate the vehicle's transverse position $k$, which corresponds to the standard value $\mathbf{B}_k^C$.

### 2.3. Identification of Gross Vehicle Weight

Assuming that a vehicle with $n$ axles drives across the bridge at speed $v$ in transverse position $i$, the strain response at measurement point $j$ can be expressed as follows:

$$\varepsilon_{ij}(t) = \sum_{i=1}^n A_i I(t - \frac{L_i}{v}), \tag{5}$$

where $A_i$ is the $i$th axle's weight and $I(t - \frac{L_i}{v})$ is the strain influence line. $L_i$ is the wheelbase. Here, it is defined as the distance between the $i$th axle and the first axle.

The strain response area of this measurement point can be expressed as follows:

$$S_{ij} = \int_{t_1}^{t_2} \varepsilon_{ij}(t)dt = \sum_{i=1}^n A_i \int_{t_1}^{t_2} I(t - \frac{L_i}{v})dt, \tag{6}$$

where $t_1$ and $t_2$ are the influence line's start point and end point, respectively, which are the same as those listed in Equation (3).

Since the area of the strain influence line for each axle is equal, the following equation can be formulated:

$$\int_{t_1}^{t_2} I(t - \frac{L_1}{v})dt = \int_{t_1}^{t_2} I(t - \frac{L_2}{v})dt = \cdots\cdots = \int_{t_1}^{t_2} I(t - \frac{L_n}{v})dt = \int_{t_1}^{t_2} I(t)dt, \tag{7}$$

We can substitute the above equation into Equation (6) to obtain the following:

$$S_{ij} = \int_{t_1}^{t_2} \varepsilon_{ij}(t)dt = \sum_{i=1}^n A_i \int_{t_1}^{t_2} I(t)dt, \tag{8}$$

Thus, the total weight of the vehicle $A_G$ can be calculated, as follows:

$$A_G = \sum_{i=1}^n A_i = \frac{S_{ij}}{\int_{t_1}^{t_2} I(t)dt}, \tag{9}$$

As seen from the above equation, the total weight of the vehicle load is only related to the strain response area of the measurement point under the vehicle load and the influence line's area.

To obtain the area of the influence line, we can measure the strain response $\varepsilon_{ij}^C$ under a calibration vehicle with known weight $A_G^C$ passing through the bridge from different

transverse positions $i$, and the influence line's area can be calculated based on the strain response area $S_{ij}^C$:

$$\int_{t_1}^{t_2} I(t)dt = \frac{S_{ij}^C}{A_G^C},$$ (10)

In the following, the strain response under an unknown vehicle load is collected, and the strain response area vector $\mathbf{B}_T = [S_{T1}, S_{T2}, \cdots, S_{T6}]$ is calculated. Based on the proposed cosine similarity *CI* index, the transverse position $k$ of the vehicle can be determined. Finally, the unknown vehicle's gross weight $A_G^T$ can be identified as follows:

$$A_G^T = \frac{S_{Tj}}{S_{kj}^C} \cdot A_G^C,$$ (11)

It can be seen from the above theoretical that one axle sensor measurement point is enough to identify the vehicle's gross weight. However, the response of the measurement point is sensitive to the transverse position of the vehicle. The strain response is small and the signal-to-noise ratio is low when the vehicle is far away from the position of the measurement point. Thus, multiple sensors were arranged in this study, and all of the identified results were averaged to decrease the influence of the noise.

## 3. Numerical Simulation

To verify the effectiveness of the proposed method, a finite element model was built on the basis of an actual orthotropic deck steel box beam bridge. The numerical simulation results were used to identify the vehicle's transverse position and gross weight.

### 3.1. Model Introduction

The numerical model dimensions refer to an actual orthotropic steel box girder bridge, whose cross-section is shown in Figure 4. Considering the significant local force characteristic, we built a local finite element model with two lanes with a width of 7.5 m in the transverse direction and three-span diaphragms with a length of 9.6 m in the longitudinal direction; the diaphragm was cut out with 0.8 m down from the deck in the vertical direction. The deck, U-rib, and diaphragm thicknesses were 14 mm, 12 mm, and 8 mm, respectively. The general commercial FEM software ANSYS was adapted to simulate the orthotropic steel box girder model. The whole model was simulated using shell elements; the material elastic modulus was E = 2.1 × 10⁵ MPa and the Poisson ratio was 0.3. Without a loss of generality, the load identification of the vehicle driving in the middle lane was mainly considered, so the mesh size of the middle lane was refined to 2 cm and the mesh size of other parts was set as 4 cm. The bottom of the diaphragms at the outermost end was simple and support-constrained, and the middle diaphragms were constrained in the vertical and transverse directions. The whole finite element model is shown in Figure 5.

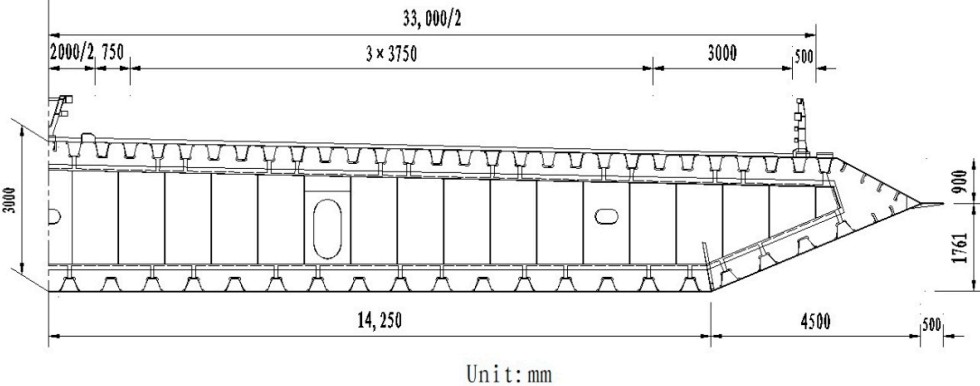

**Figure 4.** Half of the cross-section of an orthotropic deck steel box girder.

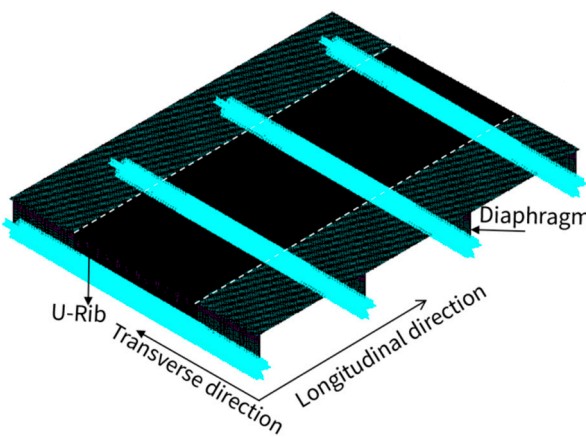

**Figure 5.** Finite element model.

*3.2. Gross Weight Identification*

(1)    Calibration of the strain response area

Before identifying the total load weight, it was necessary to calibrate the strain response area by driving a vehicle with a known weight across the bridge from different transverse positions. Here, a two-axle calibration vehicle was simulated to pass over the bridge model from the left edge position of the lane at x = −68 cm to the right edge position of the lane at x = 68 cm, every 4 cm (here, we defined it as calibration spacing *d*). The center position of the lane was defined as x = 0 cm. The calibration vehicle's front and rear axle weight was 14 kN and the wheelbase was 260 cm. By finite element calculating, the strain response $\varepsilon_{ij}$ at the position of the measurement points shown in Figure 1 could be obtained, where *i* denotes the transverse position and *j* is the number of measurement points. Thus, all of the strain responses $S_{ij}$ could be calculated, and the calibrated strain response area matrix $\mathbf{B}^C = [\mathbf{B}_1^C, \mathbf{B}_2^C, \cdots, \mathbf{B}_n^C]$ could be established. To obtain a finer location of the transverse position, the strain response area vectors $\mathbf{B}_i^C$ were refined at transverse positions every 1 cm using triple spline interpolation.

(2)    Locating the transverse position

A two-axle test vehicle was simulated to pass over the model from the transverse position x = −42 cm. The front and rear axle weights were 7 kN and 14 kN, respectively, and the wheelbase was 300 cm. The structural strain response could be calculated by FEA. Considering the influence of the noise, 10% and 20% (defined as noise variance/signal variance) Gaussian white noise signals were added to the response data. The strain response with 20% noise is shown in Figure 6.

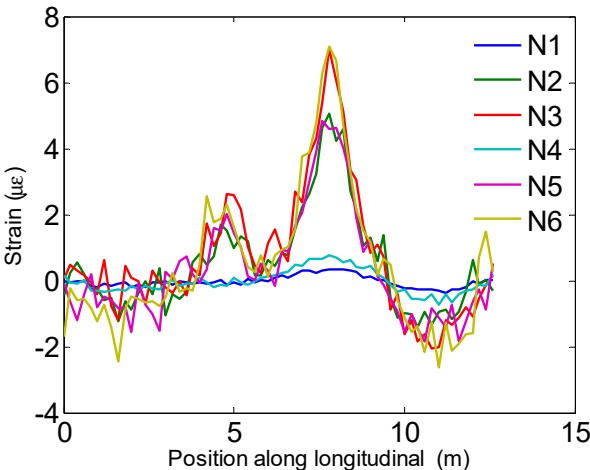

**Figure 6.** The strain response of the measurement point with 20% noise.

According to the proposed method, the cosine similarity *CI* index could be calculated based on the strain response and the calibrated strain response area matrix. The relation between the evaluated *CI* index and the transverse position is shown in Figure 7.

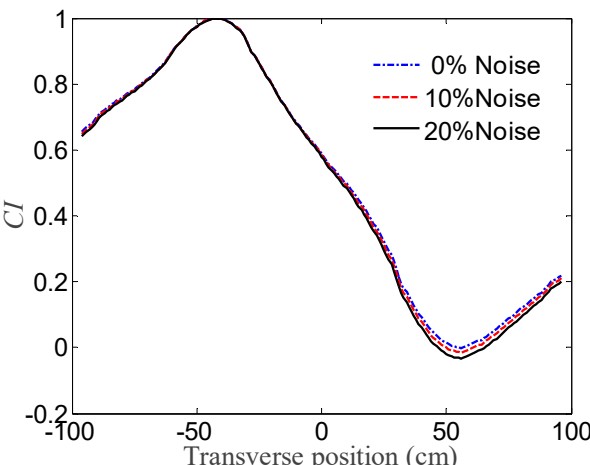

**Figure 7.** The variation curve of the *CI* index relative to the transverse position.

As seen from the figure, the cosine similarity *CI* index reached the maximum at the transverse position x = −42 cm under different noise cases, which is consistent with the practical position. Thus, the proposed index could effectively locate the transverse position with good anti-noise performance.

Theoretically speaking, the strain response is nonlinear due to the dynamic effect. The proposed *CI* index is used to calculate the degree of linear correlation between the input vectors. However, here, the strain response was not directly used as an input. Instead, the strain response area was used as an input. In the example of a two-axle vehicle crossing the model at speed of 10 m/s, the front and rear axle weights were 10 kN and 15 kN, respectively, and the wheelbase was 320 cm. The static and dynamic responses of N2 are shown in Figure 8.

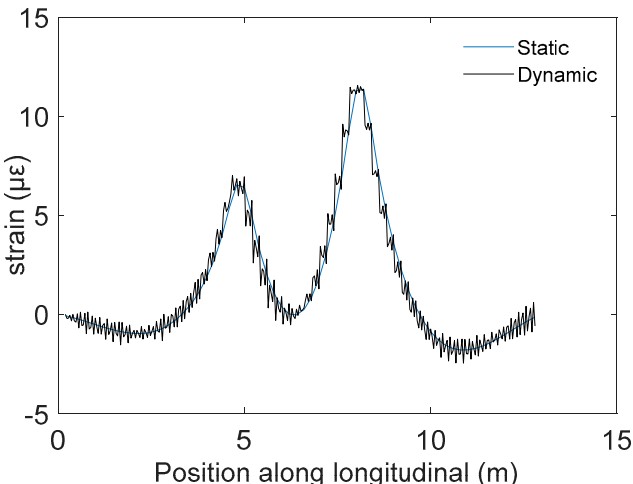

**Figure 8.** The static and dynamic response of N2.

As shown, the responses were different, albeit only slightly; the static and dynamic strain response areas were 18.2732 and 18.2752, respectively. From the above identification results of the transverse position, it also can be seen that the error was acceptable using cosine similarity.

(3)    Evaluating the gross vehicle weight

To validate the gross weight identification effect, different types of vehicles were driven across the bridge model from different transverse positions, and the corresponding strain responses were obtained using finite element analysis. A total of five types of vehicles were simulated, and the wheelbase and axle weight parameters are listed in Tables 1 and 2, respectively.

**Table 1.** Wheelbase information for different vehicles.

| Type of Vehicle | Number of Axles | D2 (m) | D3 (m) | D4 (m) | D5 (m) | D6 (m) |
|:---:|:---:|:---:|:---:|:---:|:---:|:---:|
| V1 | 2 | 2.6 | - | - | - | - |
| V2 | 3 | 3.0 | 1.4 | - | - | - |
| V3 | 4 | 2.0 | 4.0 | 1.4 | - | - |
| V4 | 5 | 3.2 | 1.4 | 6.0 | 1.4 | - |
| V5 | 6 | 3.2 | 1.4 | 7.0 | 1.4 | 1.4 |

Note: D$i$ denotes the wheelbase from the $i$th axis to the $(i-1)$th axis.

**Table 2.** Axle and gross weight for each vehicle.

| Type of Vehicle | A1 (kN) | A2 (kN) | A3 (kN) | A4 (kN) | A5 (kN) | A6 (kN) | GVW (kN) |
|:---:|:---:|:---:|:---:|:---:|:---:|:---:|:---:|
| V1 | 14 | 14 | - | - | - | - | 28 |
| V2 | 14 | 14 | 14 | - | - | - | 42 |
| V3 | 14 | 14 | 14 | 28 | - | - | 70 |
| V4 | 14 | 14 | 14 | 14 | 28 | - | 84 |
| V5 | 14 | 14 | 14 | 14 | 28 | 28 | 112 |

Note: A$i$ denotes the axis weight of the $i$th axis. GVW denotes the gross vehicle weight.

The proposed method was applied to locate the transverse position, and then we could identify the vehicle's gross weight based on the calibrated strain response area. The identified results of the transverse position (denoted as TP) and the gross vehicle weight (denoted as GVW) are shown in Table 3.

**Table 3.** Load identification of vehicles on each axle.

| Type of Vehicle | Noise Level | Practical TP (cm) | Identified TP (cm) | Absolute Error of TP (cm) | Identified GVW (kN) | Error of GVW (%) |
|:---:|:---:|:---:|:---:|:---:|:---:|:---:|
| V1 | 0% | −57 | −58 | 1 | 28.02 | 0.10 |
|  | 10% |  | −58 | 1 | 27.39 | −2.17 |
|  | 20% |  | −58 | 1 | 28.79 | 2.81 |
| V2 | 0% | 31 | 31 | 0 | 41.99 | −0.03 |
|  | 10% |  | 30 | 1 | 41.35 | −1.55 |
|  | 20% |  | 31 | 0 | 40.71 | −3.07 |
| V3 | 0% | −50 | −50 | 0 | 70.06 | 0.08 |
|  | 10% |  | −50 | 0 | 69.83 | −0.24 |
|  | 20% |  | −51 | 1 | 71.47 | 2.10 |
| V4 | 0% | 2 | 2 | 0 | 84.09 | 0.11 |
|  | 10% |  | 2 | 0 | 85.66 | 1.97 |
|  | 20% |  | 3 | 1 | 85.04 | 1.24 |
| V5 | 0% | −54 | −54 | 0 | 111.98 | −0.01 |
|  | 10% |  | −54 | 0 | 112.16 | 0.14 |
|  | 20% |  | −54 | 0 | 114.09 | 1.87 |

It can be seen that the maximal absolute identified error of the transverse position was 1 cm in all cases, and the identified result was not sensitive to the noise, which indicates that the cosine similarity index has good anti-noise performance. For the identification of the GVW, the error

increased with noise enhancement, and the maximal error was about 3% under 20% noise, which can meet the need of practical engineering applications.

### 3.3. Influence of Calibration Spacing

When we calibrated the stress response area in practical application, we needed to choose appropriate calibration spacing $d$. The smaller the parameter $d$ was, the higher the load identification accuracy that could be obtained; however, this may greatly increase the workload on site, and vice versa. In addition, it was difficult to accurately control the vehicle that travelled at a given spacing $d$ on the deck. Here, we studied the influence of different calibration spacing $d$ on identifying the GVW.

(1)  Constant calibration spacing

In the numerical simulation above, the calibration spacing was set as $d = 4$ cm, and then the strain response area vectors were refined to 1 cm using triple spline interpolation. Then, we changed the calibration spacing $d$ from 4 cm to 48 cm and established the corresponding calibrated strain response area matrix $\mathbf{B}^C$. In the following, the same two-axle test vehicle was simulated to pass over the bridge from the transverse position x = −36 cm, and the structural strain response was obtained using FEA with different noise. Based on the calibrated strain response area matrix with different calibration spacing $d$, we could locate the TP and identify the GVW of the test vehicle using the proposed method. The identified results are shown in Figures 9 and 10 below.

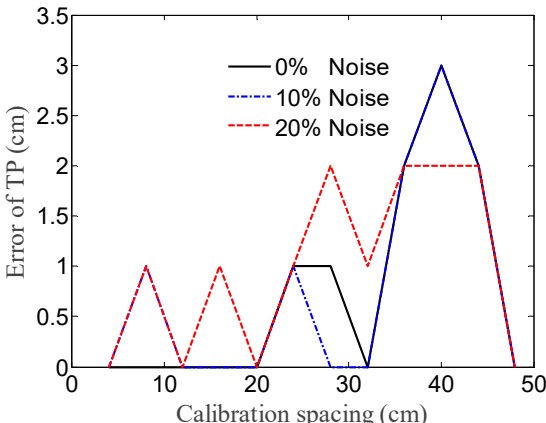

**Figure 9.** Identification error of the TP under different calibration spacing $d$.

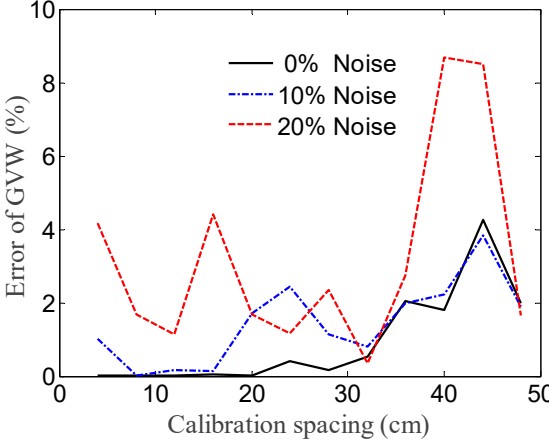

**Figure 10.** Identification error of the GVW under different calibration spacing $d$.

It can be seen that when the calibration spacing $d$ varied from 4 cm to 48 cm, the identification errors of the GVW and the TP showed an overall increasing trend. When

the calibration spacing *d* was less than 32 cm, the maximum identification error of the TP did not exceed 2 cm under all kinds of noise, and that of the GVW was below 5%. If the calibration spacing *d* exceeded 32 cm, both errors sharply increased. Thus, these results indicate that the calibration spacing should not exceed 32 cm.

(2)    Variable calibration spacing

The influence of different calibration spacing *d* on the identification of the GVW was studied, and the maximum calibration spacing was determined as 32 cm in the above section. In practice, it is difficult to let the calibration vehicle accurately pass over the bridge from the specified transverse position, so we further studied the influence of random calibration spacing *d* on the identification of the GVW. The detailed method is listed below:

① The structural strain response was extracted under the two-axle calibration vehicle passing over the bridge model at the transverse position from x = −68 cm to x = 68 cm every 4 cm;

② The strain response was randomly selected at six transverse positions and the maximum calibration spacing *d* was ensured to be 28 cm;

③ The calibrated strain response area matrix was established and the transverse position was interpolated to 1 cm;

④ The TP was located and the GVW of the test vehicle was identified, as presented in the previous section of constant calibration spacing;

⑤ Steps ② to ④ were repeated one hundred times to obtain identified results multiple times.

When the maximum calibration spacing *d* was 28 cm, the identified results of the GVW are shown in Figure 11.

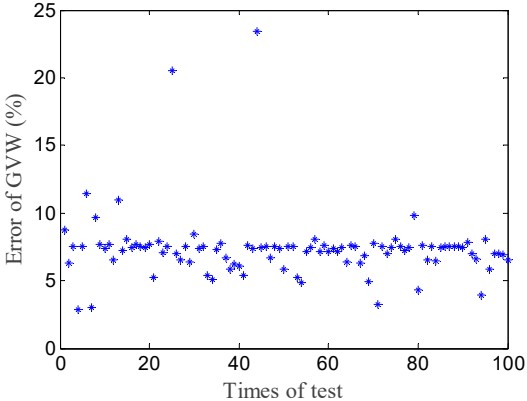

**Figure 11.** The identified errors of the GVW with 20% noise when *d* = 28 cm.

The maximum calibration spacings *d* were set as 32 cm and 36 cm, and the above steps ② to ⑤ were then repeated so that we could obtain the identified results under larger calibration spacing. The identified results are shown in Figures 12 and 13.

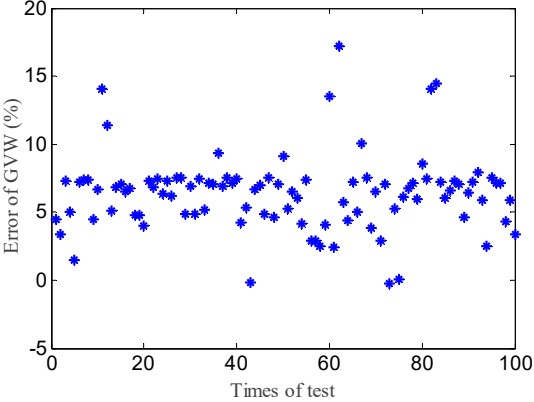

**Figure 12.** The identified errors of the GVW with 20% noise when *d* = 32 cm.

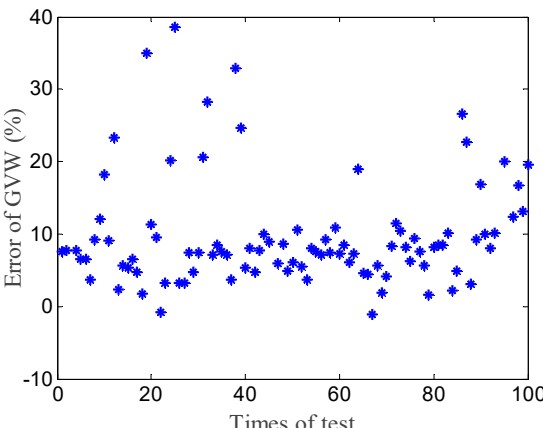

**Figure 13.** The identified errors of the GVW with 20% noise when $d = 36\,$cm.

It can be seen that when the maximum calibration spacing was no more than 28 cm, most of the identified errors in the GVW were below 7.5% under 20% noise. With an increase in maximum calibration spacing, the identified errors became larger and more scattered. This was mainly because larger calibration spacing decreased the accuracy of the strain response area vectors when interpolated to 1 cm intervals. Thus, we could let the calibration vehicle randomly pass over the bridge, and select six groups of strain response where the maximum calibration spacing $d$ was no more than 28 cm to build the strain response area matrix, which is convenient for practical engineering applications. Certainly, if the strain response of more positions with less calibration spacing was selected, better identified results could be obtained.

## 4. Experimental Test

### 4.1. Experimental Model

To further validate the proposed algorithm, a 1:6 scaled-down steel beam model was designed for testing. The model was 270 cm-long, 83 cm-wide, and 20 cm-high, with six U-ribs under the deck and six diaphragms with 54 cm spacing. A two-axle vehicle was designed to simulate the vehicle load. The vehicle dimensions were 45 cm × 40 cm × 28 cm (length, width, and height, respectively). The wheelbase, tread, and wheel width of the vehicle were 38 cm, 28 cm, and 2 cm, respectively. Moreover, the axle and gross weight could be adjusted by adding a mass block to the vehicle. The steel beam model was supported at the bottom of the first and last diaphragms. The lead and tail beam were installed at the beginning and end of the steel beam, respectively, to ensure that the vehicle could pass through the steel beam at a uniform speed. Two limit holes were set on the vehicle, and two parallel wires fixed on both ends passed through the limit holes in order to ensure that the vehicle could travel across the model in a straight line. A motor was fixed at the end of the tail beam, which was used to drag the vehicle across the steel beam. The installed test model is shown in Figure 14.

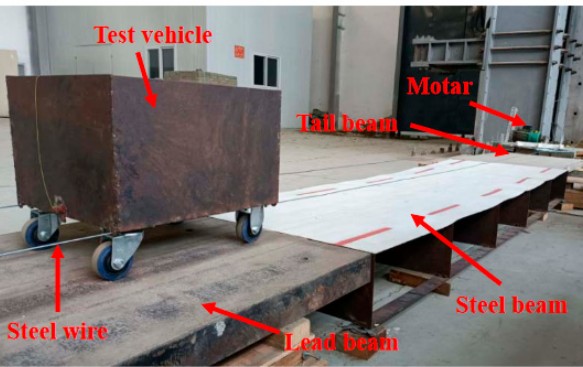

**Figure 14.** The whole experimental model.

Six strain gauges (N1 to N6) were placed along the longitudinal direction at the lower edge of every U-rib in the middle section. Two strain gauges (N7 and N8) were installed at the lower edge of the same U-rib in the adjacent section. The detailed dimensions of the model and the layout of measurement points are shown in Figure 15.

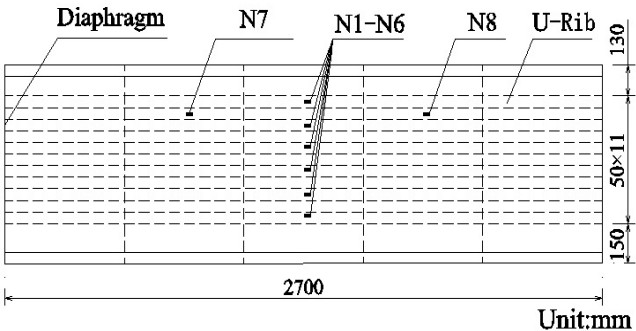

**Figure 15.** The layout of the strain measurement points.

*4.2. Load Calibration*

Initially, some mass blocks were added to the experimental trolley, and the total weight was 87.2 kg. The motor pulled the vehicle across the steel beam at different transverse positions from x = −21.5 cm to x = 16.5 cm. Here, the transverse center position on the deck was defined as x = 0 cm. A total of 39 transverse positions were tested, and the corresponding strain responses at measurement points N1~N8 were obtained using a sampling frequency of 50 Hz. Figure 16 shows the stress responses of measurement points when the vehicle passed over the model at the transverse position x = −1.5 cm (the strain was transformed into stress for the convenience of the display).

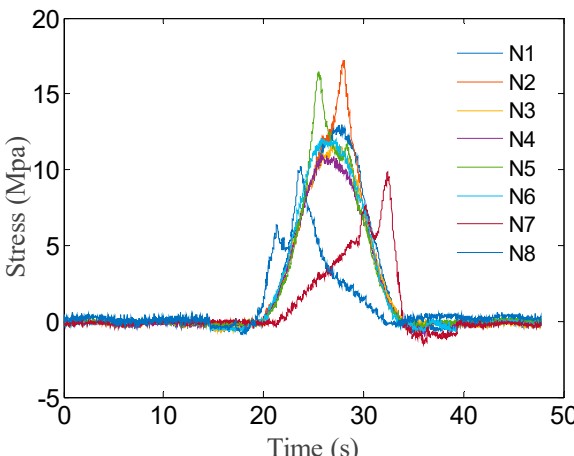

**Figure 16.** Stress response of measurement points N1~N8 for TP x = −1.5 cm.

The stress response data of measurement points N7 and N8 were processed using the proposed correlation algorithm, and the vehicle's speed was subsequently identified as $v = 12.05 \, \text{cm/s}$.

Next, the strain response areas of measurement points N1 to N6 were calculated based on the estimated vehicle's speed. The calibrated strain response area matrix was then built using the proposed method.

*4.3. Load Identification*

In the following, the vehicle's weight was adjusted by adding different mass blocks, and experimental vehicles were driven through the steel beam model from different trans-

verse positions. The strain response under multiple test cases could be collected for load identification purposes.

Based on the calibrated strain response area matrix, the proposed *CI* index was calculated to estimate the transverse position under different cases, and then the GVW could be identified. For the example of case 4, the variation curve of the evaluated *CI* index with the transverse position is shown in Figure 17.

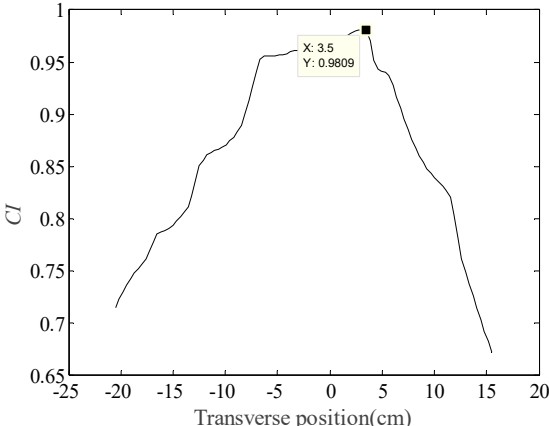

**Figure 17.** The variation curve of the *CI* index with the transverse position under case 4.

It can be seen that the *CI* index reached the maximum 0.9809 at the transverse position x = 3.5 cm. The practical transverse position was 3.7 cm and its error was acceptable. Thus, the index could effectively locate the transverse position.

All the identified results are listed in Table 4.

**Table 4.** Identified results of the TP and the GVW.

| Cases | Practical TP (cm) | Identified TP (cm) | Error of TP (cm) | Identified GVW (kg) | Error of GVW (%) |
|---|---|---|---|---|---|
| 1 | −15.5 | −15.5 | 0 | 96.9 | 3.37 |
| 2 | −6.5 | −5.5 | 1 | 96.9 | 1.2 |
| 3 | −3 | −2.5 | 0.5 | 96.9 | 3.83 |
| 4 | 3.7 | 3.5 | 0.2 | 96.9 | 4.69 |
| 5 | −16.3 | −16.5 | 0.2 | 106.6 | 2.6 |
| 6 | −16.5 | −16.5 | 0 | 106.6 | 2.32 |
| 7 | 4 | 3.5 | 0.5 | 106.6 | 4.19 |
| 8 | 11.7 | 11.5 | 0.2 | 106.6 | 2.41 |

As seen from Table 4, the maximum identified error of the transverse position was 1 cm based on the proposed *CI* index; most of them were less than 0.5 and the maximum identified error of the GVW was 4.69%. This indicates that the proposed method can be effectively used to identify the transverse position and the gross weight, with good anti-noise performance.

### 4.4. Effect of Calibration Spacing

To study the effect of the transverse calibration spacing *d* on the identification of the TP and the GVW, the stress response areas were calculated from the measured data with the *d* being 1~8 cm. Then, the stress response area matrix was obtained by interpolating to 1 cm intervals. A two-axle test vehicle with a weight of 96.9 kg was pulled across the model at the transverse position x = −15.5 cm. The proposed method was used to identify the TP and the GVW. The identified results are shown in Figures 18 and 19.

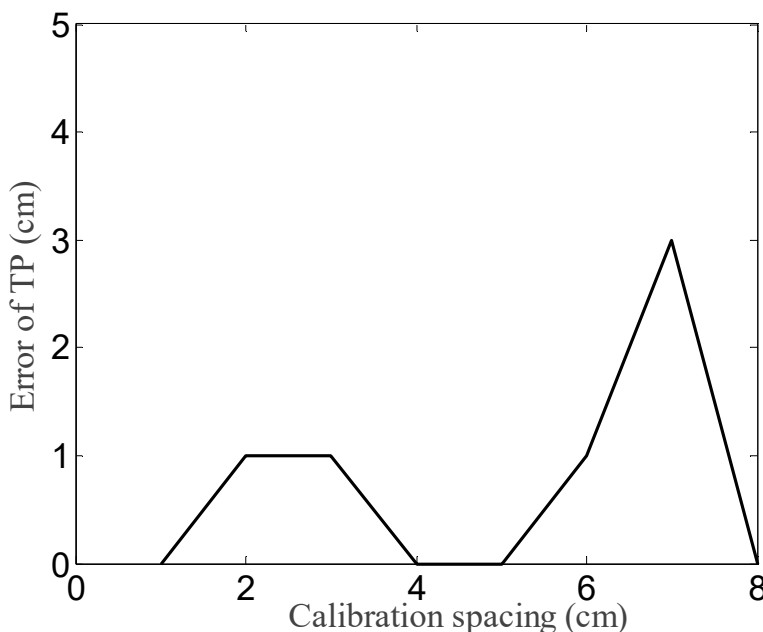

**Figure 18.** Identified error of the TP under different calibration spacing *d*.

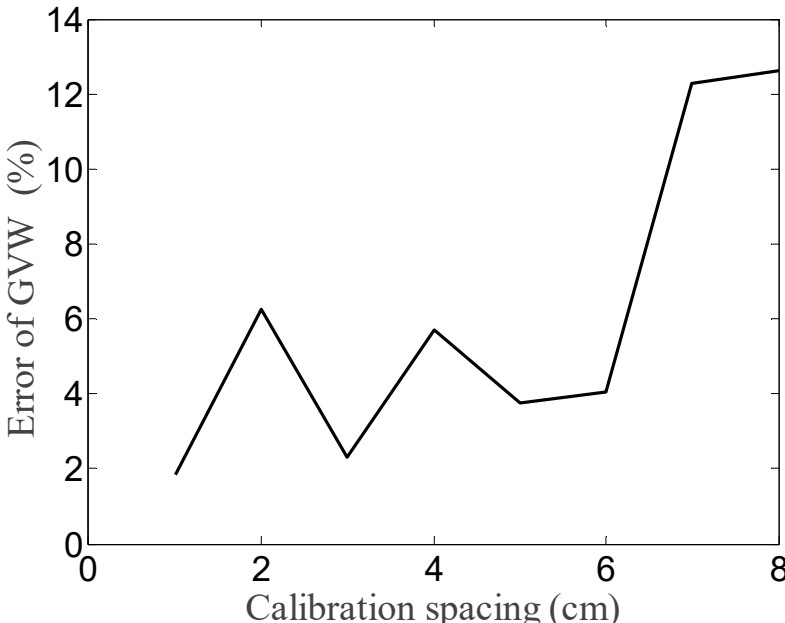

**Figure 19.** Identified error of the GVW under different calibration spacing *d*.

It can be seen that when the calibration spacing was less than 6 cm, the identified error of the TP was about 1 cm, and that of the GVW was below 7%. When the calibration spacing continued to increase, the fluctuation of identified TP error became larger and the identified GVW error rapidly increased. Thus, the appropriate calibration spacing *d* was no more than 6 cm, which is consistent with the numerical simulation, considering that the experimental test was a 1:6 scaled-down model.

## 5. Conclusions

With the rapid development of the economy, traffic overload often occurs. Accurately identifying the vehicle loads has important theoretical significance and engineering application value for structural health monitoring and safety assessment. This study proposed a

vehicle load identification technique for orthotropic steel box beam bridges based on the strain response area. A numerical simulation and experimental test were carried out to validate the effectiveness and anti-noise performance of the proposed method. From the identified results, the following conclusions can be drawn:

The cross-correlation function between the measurement points on the same U-rib can be effectively used to evaluate the vehicle's speed.

The maximum identified error of the TP is 1 cm based on the cosine similarity *CI* index in the numerical simulation and experimental tests. This verifies the effectiveness and good anti-noise performance of the *CI* index.

In practical engineering applications, we can let the calibration vehicle randomly cross the deck at different transverse positions, and then select the strain response at six or more transverse positions to build a calibrated strain area matrix, only ensuring that the maximum calibration spacing *d* is less than 28 cm.

The proposed method can effectively identify the gross vehicle weight with good anti-noise performance. It only uses the strain response of the measurement point installed at the lower edge of the U-rib of the orthotropic deck steel box beam bridge, which does not damage the bridge deck, does not affect the traffic, and has good economic benefits compared to expensive road-based dynamic weighing systems.

**Author Contributions:** Conceptualization, C.W.; Methodology, T.-Y.Q.; Software, Z.-S.Z.; Validation, T.-Y.Q.; Formal analysis, J.-H.Z.; Investigation, T.-Y.Q.; Data curation, Z.-S.Z.; Writing—original draft, J.-H.Z.; Visualization, T.-Y.Q.; Supervision, C.W.; Project administration, C.W. All authors have read and agreed to the published version of the manuscript.

**Funding:** This research is supported by the National Natural Science Foundation of China (Grant No. 51408250) and the Hubei University of Technology Postgraduate Innovative Talent Training Program (Grant No. Univ. 2022054).

**Institutional Review Board Statement:** Not applicable.

**Informed Consent Statement:** Not applicable.

**Data Availability Statement:** All necessary data is given in this article.

**Acknowledgments:** This research was financially supported by the National Natural Science Foundation of China (grant no. 51408250) and the Hubei University of Technology Postgraduate Innovative Talent Training Program (grant no. Univ. 2022054). The authors would also like to thank the reviewers for their suggestions which permit to improve this manuscript.

**Conflicts of Interest:** The authors declare no conflict of interest.

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
