# Peer review of "Vehicle Load Identification on Orthotropic Steel Box Beam Bridge Based on the Strain Response Area"

_applsci, doi:10.3390/app122312394_

Round 1

Reviewer 1 Report

This article introduces a strategy to identify the vehicle load on the orthotropic steel box beam. There are some problems in its current form and some revisions that shall be made in order to improve the quality of this article. The novelty of the proposed method is limited and the technical contents supporting the conclusions are inadequate.

Followings are some of my comments and suggestions for this article:

1.     The figures (e.g., Fig. 2,7,8) should obey the relevant requirements of MDPI.

2.     Please add a punctuation after each equation.

3.     In subsection 3.1, the authors should indicate which platform has been used to establish the FE model, such as ansys or matlab. In addition, the dynamic load should be used and the corresponding speed should be provided in this section.

4.     In subsection 3.2, the “cosine similarity index” (Eq. 4) is used to detect the “transverse position” in this manuscript. However, this function is to calculate the degree of linear correlation between the input vectors. Theoretically, the variation of the input vector (i.e., strain response) is nonlinear due to the dynamic responses. If the error is acceptable or the variation is still linear, the authors should conduct some further investigation to support this statement.

5.     In subsection 4.2, Fig.14. Please explain why the stress peaks of N2 and N5 is at different x coordinates (i.e., time) in contrast to that of N1 and N6, since the car is planned to arrive at N1~N6 at the same time, as the manuscript says “Two limit holes were set on the vehicle, and … to ensure that the vehicle could travel on the model in a straight line” (page 12, line 337).

6.     In subsection 4.2, line 363. The vehicle’s speed is calculated by the “cross-correlation function” (Eqs. 1 and 2). However, if the conditions for this measurement are as simple as that discussed in this section (such as the parallel moving and the one-car excitation), the difference between x coordinates for the peak of N7 and N8 in Fig. 14 is enough to calculate the speed. What is the innovation of the “cross-correlation function”? The authors are advised to have a further discussion to highlight the innovation of the proposed method.

Reviewer 2 Report

The paper focuses on the problem of accurately tracking the vehicle position and load estimation using the analysis of the strain response on the specific study case of Orthotropic Steel Box Beam Bridges. The authors propose an approach that is based on the use of the cross-correlation function of strain response between different measurement points to first evaluate vehicle speed. Then, they use a cosine similarity index to locate the transverse position of the vehicle. Finally, the unknown vehicle load can be identified based on the calibrated strain response area matrix The methodology is then validated using both a numerical and laboratory experiment. The manuscript is original, and the scientific method is valid. However, the presentation of the methodology and the results is weak and not well-organized. I recommend introducing and addressing the following changes.

In the abstract, the authors state:

1.       Some strain gauges were installed at the lower edge of the U-rib to measure the strain response when the vehicles crossed the deck. It is not clear what they are referring to. If this is a theoretical case scenario or one of the numerical and experimental study cases, they refer to it later on in the abstract.

2.      The bridge represented in Figure 1 has no units. The reader understands what’s going on only several paragraphs later. The authors should organize better their information flow.

3.      The authors should add a small flowchart explaining the different stages of the methodology, highlighting if there are any parameters that need to be set a priori in the method.

4.      In Figure 3 the authors should add the proper important dimensions of the section

5.      There is an error in Table 2 or Table 3 for vehicle type V1. The error percentages on the GVW do not correspond to the correct ones if you do the calculations considering what is in those two tables.

6.      The authors should carefully revise the paper in terms of grammar. In figure 7 the authors write calibrattion with 2 tt

7.      The authors should add the variation curve of the index CI relative to the transverse position also for the experimental study case

8.      The authors should compare their results with those obtained from an alternative approach listed by the authors in the state-of-the-art section of the introduction to further validate their approach

9.      The authors should introduce a small section where they discuss some of the current computer vision-based applications of accurate vehicle tracking and load estimation strategies to further highlight the importance of the proposed strategy. I recommend that the authors introduce this section and talk about the advancement in traffic load distribution and damage detection approaches that are based on accurate tracking of the vehicle. Among the most recent works on the topic, there are:

One of the most recent articles on traffic load distribution relates to the YOLO application. The authors should definitely check out this work also from a state-of-the-art perspective. [1] Ge, L., Dan, D., & Li, H. (2020). An accurate and robust monitoring method of full‐bridge traffic load distribution based on YOLO‐v3 machine vision. Structural Control and Health Monitoring, 27(12), e2636.

There are new and interesting applications and use of vehicle tracking strategies for damage detection and model updating, as shown in [2] Martini, A., Tronci, E. M., Feng, M. Q., & Leung, R. Y. (2022). A computer vision-based method for bridge model updating using displacement influence lines. Engineering Structures, 259, 114129.

Round 2

Reviewer 1 Report

Quality of the manuscript has been improved to some extent.

Reviewer 2 Report

no further changes needed
